# Single-Option P300-BCI Performance Is Affected by Visual Stimulation Conditions

**DOI:** 10.3390/s20247198

**Published:** 2020-12-16

**Authors:** Juan David Chailloux Peguero, Omar Mendoza-Montoya, Javier M. Antelis

**Affiliations:** Tecnologico de Monterrey, School of Engineering and Science, Monterrey, NL 64849, Mexico; omendoza83@tec.mx (O.M.-M.); mauricio.antelis@tec.mx (J.M.A.)

**Keywords:** P300 BCI, performance assessment, visual stimuli paradigm

## Abstract

The P300 paradigm is one of the most promising techniques for its robustness and reliability in Brain-Computer Interface (BCI) applications, but it is not exempt from shortcomings. The present work studied single-trial classification effectiveness in distinguishing between target and non-target responses considering two conditions of visual stimulation and the variation of the number of symbols presented to the user in a single-option visual frame. In addition, we also investigated the relationship between the classification results of target and non-target events when training and testing the machine-learning model with datasets containing different stimulation conditions and different number of symbols. To this end, we designed a P300 experimental protocol considering, as conditions of stimulation: the color highlighting or the superimposing of a cartoon face and from four to nine options. These experiments were carried out with 19 healthy subjects in 3 sessions. The results showed that the Event-Related Potentials (ERP) responses and the classification accuracy are stronger with cartoon faces as stimulus type and similar irrespective of the amount of options. In addition, the classification performance is reduced when using datasets with different type of stimulus, but it is similar when using datasets with different the number of symbols. These results have a special connotation for the design of systems, in which it is intended to elicit higher levels of evoked potentials and, at the same time, optimize training time.

## 1. Introduction

Brain-Computer Interfaces (BCI) were first proposed almost fifty years ago as an alternative output pathway to allow people communication and control of external devices without performing muscular activity [1]. Since then, this technology has evolved considerably and nowadays the main applications are found in clinical environments. Areas, such as neuro-rehabilitation for patients with neurodegenerative diseases [2,3,4,5,6], as well as assistive technologies for people with motor impairments [7,8,9,10], have had the widest presence. However, BCI’s applications have now transcended the clinical environment [11,12]. BCIs commonly rely on the non-invasive electroencephalogram (EEG) technique to record the brain activity and on a mental task to generate control signals, such as P300 Event-Related Potentials (ERP) or Steady-State Visually Evoked Potentials (SSVEP). Today, scientific efforts to successfully incorporate BCIs into daily life activities by end users are mainly focused towards improvements in performance [13], either by reduction of calibration time [14] or development of novel stimuli presentation strategies [15], among other aspects.

Notably, EEG-based P300-BCIs have been the focus of numerous investigations since they stand out for its relative easiness and also because they have shown great potential in different applications [16]. These BCIs are based on the "oddball paradigm" where a rare (i.e., target) stimulus is presented among other frequent but irrelevant (i.e., non-target) stimuli. The set of stimuli are commonly presented on a screen, and they are intensified or highlighted in random order, so that, communication is achieved by focusing attention on a target stimulus and silently counting the number of times it flashes, while ignoring the other non-target stimuli [17]. This process elicits the P300 potential solely in the target stimulus, a positive deflection in the EEG signals with a latency of around 300 ms after stimulus presentation that is associated with cognitive processes, such as attention and decision-making [18]. Hence, a machine-learning model is needed to detect the presence of the P300 potential by discriminating between target and non-target events from the EEG signals and, in consequence, to identify the stimulus the user is attending to.

One important aspect of P300-BCIs is the presentation of the visual stimuli which is carried out through a Graphical User Interface (GUI) [19]. This is because the characteristics of the visual stimuli provide the framework to interact with the system, to select targets, and, consequently, to evoke P300 responses. Indeed, parameters, such as shape, size, color, and type of flash, may enhance or diminish the difference between target and non-target responses and, thus, may influence the performance of the machine-learning model [20,21,22]. Therefore, an alternative to improve the accuracy and reliability of P300-BCIs is to employ stimuli properties that evoke stronger P300 responses [23] and that might also elicit other ERP components that occur before or after the P300 [24]. Such components may be the P100, N170, and N400 deflections which are associated with working memory, visual processing, and other cognitive functions triggered during the stimulation. In this regard, previous studies have shown that P300-BCI employing facial-type flashes elicit visual-related N170 and N400 ERP components, in addition to the P300 components [25,26,27].

Some studies have investigated the effect that different stimulation conditions, in particular the type of flash, have on the ERP components and/or on the discrimination between target and non-target responses. For instance, grey semi-transparent familiar faces have shown to evoke higher ERP components and to provide higher classification accuracy than those achieved with standard flashes [28]. Faces of relatives and famous people, and standard flashes have been compared, and the results have indicated better performance using face stimuli instead of the classical flashes [29], though no differences were found between the types of faces [30]. Faces and standard flashes of different sizes (large or small) have also been compared with results showing waveform differences though no significant difference in BCI performance [31]. Pictures of locations and graspable tools as flashes have also been studied, and the results have shown unique and discriminable brain responses that can be used to improve classification accuracy [32]. In addition, semi-transparent colored unfamiliar face patterns have been proposed where red semi-transparent faces have provided the highest BCI performance [33].

Despite the fact that various studies have demonstrated that human faces elicit stronger ERP components and therefore provide higher BCI performance than those provided by standard flashes, the face-type stimuli can lead to copyright infringements that limits its use and implementation [34]. Due to this, the use of cartoon faces is advisable since they do not present this limitation. Previous works have studied this aspect, and they have reported no significant differences in ERP components and in BCI performance between cartoon faces and human faces [34,35,36]. However, there are no works that have studied differences between the standard flash and cartoon faces, without taking into account the semantic connotation of facial expression [37,38]. Likewise, another property that has not been explored is the number of symbols that are intensified (flashed) during the presentation of the visual stimuli in target selection BCI applications. Indeed, regardless of the visual stimulation paradigm (e.g., row-column, single-option, checkerboard, region-based with levels), the number of options can range from 1 [36,39] up to 24 [40,41,42,43]. Hence, this is a critical property since it imposes the number of flashes required to activate all options presented to the user and, thus, influences the time required for calibration and online selections. Another aspect that has gained interest in BCI research is the possibility of relying on machine learning models in which performance is minimally affected by the variation in the characteristics used to infer the different classes. This usually occurs when there are variations at the session and subject level. This issue is very important and can occur in applications where users are losing physical and cognitive faculties from one session to another due to degenerative diseases, such as Amyotrophic Lateral Sclerosis, Parkinson’s, Alzheimer’s, etc.

Motivated by the lack of studies addressing these issues, the purpose of this work was to assess the effects on the ERP waveform and on the single-trial classification between target and non-target responses. We intend to do it considering two conditions of the visual stimuli: the visual stimulation (standard flashes or cartoon faces) and the number of symbols presented to the users (from four to nine). To this end, we designed a P300 experimental protocol which was carried out with 19 healthy participants in 3 sessions. The results revealed that the ERP responses and the classification performance are stronger with cartoon faces than with standard flashes and similar irrespective of the amount of options presented to the user. This is significant because sets standards in P300 BCI’s visual interface design for target selection applications. The results also showed that the classification performance of training and testing the machine learning model is reduced when using datasets containing different type of stimulus, but it is similar, regardless of the number of symbols. This is important because it would yield answers about the optimal number of symbols required to train the interface. Under these premises, our contribution is addressed to the optimization of BCI performance from establishing design parameters in the *single-option* visual stimulation paradigm, where only one element of those presented in the visual scheme flashes at a time and in which applications can be found in navigation systems [44] or remote control of devices, in addition to the clinical environment. Finally, the recorded EEG data is freely available to anyone interested in the study, evaluation, and implementation of signal processing and machine-learning algorithms for P300-BCI. The rest of the manuscript is organized as follows: Section 2 describes the experimental protocol and the methodology; Section 3 presents the results; Section 4 discuss the results and presents the conclusions.

## 2. Materials and Methods

Here, we describe the P300 paradigm experiments carried out to record EEG signals from healthy volunteers; the collected EEG datasets containing target and non-target responses to visual stimuli with different schemes (stimulation condition and number of symbols); and the data analysis methodology carried out to study the effects of those scenarios on the classification between target and non-target responses.

### 2.1. Experimental Protocol

The experiments were conducted in an acoustically isolated room where only the participant and the experimenter were present. Participants were seated in a comfortable chair in front of a computer screen (17″, LED Technology, Dell, Round Rock, TX, USA). On this screen, a Graphical User Interface (GUI) showed the stimuli (distributed uniformly on the screen) and an instruction box that guided the participants on the execution of the experiment (presented at the bottom of the screen). The GUI allows us to vary the parameters of the stimuli, such as stimulus shape, size and color, the number of symbols, and the flash condition, among others. Figure 1a shows a snapshot of the experimental setup with a participant, the computer screen displaying the GUI with a set of stimuli and the instruction box, and the EEG recording system.

A P300 experiment was carried out in several consecutive blocks. The temporal timing of each block is depicted in Figure 1b and consisted of the following five phases:*Fixation*. A cross symbol is shown for 2 s in the information box, which indicates to be prepared and relaxed.*Target Presentation*. One of the symbols on the screen is randomly highlighted with a blue background for 2 s, and the same symbol is also shown in the information box. This indicates the participant the location of the screen and the particular symbol that they have to focus their attention during the subsequent *"Stimulation"* phase.*Preparation*. No stimuli is highlighted or shown in the information box. This lasts one second and indicates to the participant that he or she must be ready for the upcoming phase.*Stimulation*. The stimuli flash randomly, one at a time, with both no-repeat and equiprobable selection constraint. The participants are asked to silently count each time the target stimuli flashes, while ignoring when the other stimuli flash. Each flash consists of highlighting a stimulus for 75 ms followed by other 75 ms without any highlighting. This phase lasts around 30 s, which slightly varies, according to the number of symbols.*Rest*. None of the symbols are highlighted, and the text “Rest” is presented in the information box. This instructed the participants to rest from the experiment for 5 s.

The duration of this block is about 40 s, and several consecutive blocks are repeated until the target stimulus flashes at least 280 times. This number of target events was chosen to obtain a sufficient number of instances that ensures significant classification rates between target and non-target [7]. The next experimental segment, which consisted of the variation in the number of symbols, also comprised the five phases previously described. The approximate overall time of a session was about fifty minutes. Note that, for each block, the target stimulus is different as it is randomly selected in the *Target Presentation* phase. Overall, if the number of symbols displayed in the GUI is $N_{stimuli}$ and the target stimulus flashes $N_{target}$ times, then the other non-target stimuli flashes $N_{nontarget}=N_{target}\cdot \left(N_{stimuli}-1\right)$ times. The execution of the experiment is fully controlled by an in-house software (SW) implemented in C++ that manages the operation of the GUI and the simultaneous acquisition of the EEG signals, along with marks that indicate the initiation of each phase, the target element, and the presentation of each of the symbols [7].

This P300 experiment was carried out by changing visual stimuli schemes: the flash condition and the number of symbols. Two conditions of visual scheme were considered: standard flash based on green-highlighting the stimulus (SF) and superimposing a yellow smiling cartoon face (CF). Figure 2a shows real screenshots of the GUI with the two types of flash. The number of symbols were varied from four to nine and they were evenly distributed on the screen. Figure 2b shows real screenshots with the different configurations for the different number of symbols. Note that the shape of the stimuli were arrow symbols (if the stimulus location is on the periphery of the screen) pointing out of the screen and/or an octagon with the text “STOP” (if the stimulus location is on the center of the screen). The other stimulus parameters, such as the shape, size, color, brightness, and transparency, were kept constant.

The experimental protocol consisted of the following procedure: Participants first conducted the P300 experiment independently with the two type of stimulation conditions (SF and CF) but only with 5 blinking symbols. The order of these two visual schemes were randomly chosen. The classification rate between target and non-target of these two conditions were computed using the procedure described in the subsequent Section 2.5 and Section 2.6. The stimulation condition that provided the higher classification rate was selected and used in the subsequent experiments. This selection was carried out individually for each participant. Participants then conducted the P300 experiment with the selected type of flash but with 4, 6, 7, 8, and 9 symbols. The order of these experiments was also random. Therefore, each participant performed in total seven P300 experiments. To avoid tiredness and boredom, participants were encouraged to rest as long as needed between P300 experiments. This procedure was carried out in three experimental sessions, separated by a maximum one week. The maximum number of days between the three sessions was 21 days, while the minimum was 10 days. The order of the stimulation condition (SF and CF with 5 symbols) and of the number of symbols (4, 6, 7, 8, and 9) was also random across the sessions.

### 2.2. EEG Data Acquisition

EEG signals were recorded from 8 scalp locations using a g.SCARABEO Ag/AgCl active biopotential electrodes system and a g.USBamp biosignal amplifier (g.tec medical engineering GmbH, Schiedlberg, Austria). The EEG electrode positions were Fz, Cz, P3, Pz, P4, PO7, PO8, and Oz, according to the 10–20 international system. These positions were employed because they are the standard scalp locations to record the P300 evoked responses [45,46,47]. The ground was placed to the AFz position and referenced to the right earlobe with an active Ag/AgCl electrode. EEG signals were recorded at a sampling frequency of 256 Hz, power-line notch-filtered and band-pass filtered from 0.5 to 60 Hz. The electrode impedance was checked and kept below 5 kΩ. This was carried out before the initiation of each P300 experiment using the g.Recorder software.

### 2.3. Participants

Nineteen healthy volunteers (11 males and 8 females) with age range from 19 to 33 years (mean 25) were recruited to participate in the study. Enrollment in the study was completely free, and no compensation was given to the recruited participants. All participants had normal or corrected-to-normal vision and also had no previous experience as BCI users. Prior to the initiation of the experimental sessions, participants were duly instructed about the nature and goals of the research, and they were instructed about the correct execution of the experiment. The experimental procedure and the study’s consent form was approved by our institution ethics committee and met the standards of the Helsinki Declaration. All participants voluntarily signed consent form and provided written authorization to take video and picture recordings.

### 2.4. Dataset Description

Each participant (sub-01 to sub-019) performed 3 experimental sessions (ses-01 to ses-03) and each session consisted of seven P300 experiments or data files. The seven data files for each session are: one for stimulation type standard flash or SF with 5 symbols, one for stimulation type cartoon face, or CF, with 5 symbols, and five for 4, 6, 7, 8, and 9 number of symbols with the stimulation condition that provided the greater classification accuracy with 5 symbols. The raw EEG signals along with a detailed description of the recorded data (see Appendix A) are freely available and can be accessed through the online site https://openneuro.org/datasets/ds003190. The datasets are formatted according to the Brain Imaging Data Structure (BIDS) standard [48]. The database for this study is also available on request to the corresponding author.

### 2.5. Single-Trial Classification

#### 2.5.1. Data Preparation and Pre-Processing

Recorded EEG data of each P300 experiment was independently subjected to the following pre-processing steps. First, EEG epochs were extracted from −0.2 to 1.0 s relative to the time of the stimulus presentation and they were labelled as target or non-target according to whether the participant was attending or not the stimulus. Here, the number of EEG epochs for the target condition is $N_{target}$ (which is at least 280 and varies slightly according to the number of symbols) and the number of EEG epochs for the non-target condition is $N_{nontarget}=N_{target}\cdot \left(N_{stimuli}-1\right)$, where $N_{stimuli}$ is the number of symbols presented on the GUI. Afterwards, the following exclusion criteria were applied to identify and discard noisy epochs: (i) Peak-to-peak amplitude greater than 200 μV; (ii) Standard deviation amplitude greater than 50 μV; and (iii) Power ratio between the frequency bands [20–40] Hz and [4–40] Hz greater than 0.5 [7]. EEG epochs with at least one electrode fulfilling any of these criteria were discarded and not used in the subsequent analyses. Accepted epochs were then band-pass filtered from 4 to 14 Hz using a Finite Impulse Response (FIR) digital filter. These data preparation and pre-processing steps lead to the cleaned dataset (for each P300 experiment) ${\left\{{\mathrm{EEG}}_{i},y_{i}\right\}}_{i=1}^{N_{t}}$, where ${\mathrm{EEG}}_{i}\in R^{N_{s}\times N_{e}}$ is the EEG activity of the *i*-th epoch, $y_{i}\in \left\{target,nontarget\right\}$ indicates whether the epoch belongs to the target or non-target condition, $N_{t}$ is the number of epochs, $N_{s}$ is the number of samples, and $N_{e}$ is the number of electrodes. Cleaned datasets were employed to compute the ERP waveform of each channel by simply computing the across-all-epochs average separately for the target and non-target conditions.

#### 2.5.2. Feature Extraction

There are several feature extraction methods suitable in the field of BCIs [49,50,51]. In our proposal, features were computed using spatial filters based on the Canonical Correlation Analysis (CCA) technique, which measures the inter-relation between two sets of random observations $P\in R^{T\times N}$ and $Q\in R^{T\times M}$, where *T* is the number of observations, and *N* and *M* are the number of variables in P and Q, respectively. CCA seeks the linear combinations $p=Pw_{p}$ and $q=Qw_{q}$ that maximize the so-called canonical correlation ρ between them. Hence, the weight vectors $w_{p}\in R^{N\times 1}$ and $w_{q}\in R^{M\times 1}$ are found by solving:
(1)$$\rho =\max_{w_{p},w_{q}}corr\left(p,q\right),$$
which can be rewritten as the following optimization problem:
(2)$$\rho =\max_{w_{p},w_{q}}\frac{w_{p}^{⊤}C_{pq}w_{q}}{\sqrt{w_{p}^{⊤}C_{pp}w_{p}w_{q}^{⊤}C_{qq}w_{q}}},$$
where $C_{pq}$ is the cross-covariance matrix, and $C_{pp}$ and $C_{qq}$ are the auto-covariance matrices for P and Q, respectively. The solution to this problem is obtained by solving a generalized eigenvalue problem [52], from which the weight vector $w_{p}$ is an eigenvector of $C_{pp}^{-1}C_{pq}C_{qq}^{-1}C_{qp}$, whereas the weight vector $w_{q}$ is an eigenvector of $C_{qq}^{-1}C_{qp}C_{pp}^{-1}C_{pq}$. It follows that several consecutive eigenvectors can be selected in descending order according to the eigenvalues to construct the spatial filter matrices $W_{p}=\left[w_{p}^{1},w_{p}^{2},\cdots ,w_{p}^{N_{sf}}\right]$ and $W_{q}=\left[w_{q}^{1},w_{q}^{2},\cdots ,w_{q}^{M_{sf}}\right]$, where $N_{sf}\leq N$ and $M_{sf}\leq M$ are the number of selected weight vectors or filters. From here, the spatial filtered (i.e., projected) data for P is $P^{sf}=PW_{p}$, while, for Q, it is $Q^{sf}=QW_{q}$.

Given a training dataset ${\left\{{\mathrm{EEG}}_{i}^{train},y_{i}^{train}\right\}}_{i=1}^{N_{t}}$, the feature extraction procedure based on CCA spatial filtering is applied as follows. First, all epochs are trimmed from 0 to 0.8 s and decimated by a factor of 4, yielding to ${\left\{X_{i},y_{i}\right\}}_{i=1}^{N_{t}}$ where $X\in R^{N_{d}\times N_{e}}$, and $N_{d}$ is the new number of reduced samples. Epochs from the target class are then selected to obtain the dataset ${\left\{X_{i}^{target}\right\}}_{i=1}^{N_{target}}$ where $N_{target}$ is the total number of epochs of the target event. The average is computed to obtain ${\bar{X}}^{target}$, which is replicated $N_{target}$ times to obtain the dataset ${\left\{{\bar{X}}_{i}^{target}\right\}}_{i=1}^{N_{target}}$. Subsequently, these two datasets of target epochs and replicated averaged target epochs are re-shaped to obtain the same-size 2D matrices $X^{\prime }\in R^{\left(N_{target}\cdot N_{d}\right)\times N_{e}}$ and ${\bar{X}}^{\prime }\in R^{\left(N_{target}\cdot N_{d}\right)\times N_{e}}$. The CCA analysis described above is then applied to these two matrices to obtain the spatial filter $W_{x^{\prime }}=\left[w_{x^{\prime }}^{1},w_{x^{\prime }}^{2},\cdots ,w_{x^{\prime }}^{N_{sf}}\right]$. Finally, given a trimmed and decimated epoch, the spatial filtered data is computed as $X^{sf}=XW_{x^{\prime }}$, which is concatenated to obtain the feature vector $x\in R^{1\times (N_{d}\cdot N_{sf})}$. Note that $W_{x^{\prime }}$ is computed exclusively from training data and provides a lower dimensional representation if $N_{sf}<N_{e}$. Here, we employed $N_{sf}=3$ spatial filters as this is sufficient to capture all the underlying activity of the EEG [7,53].

#### 2.5.3. Classifier

To discriminate between target and non-target responses, Linear Discriminant Analysis (LDA) with regularized (shrinkage) covariance was used as classification model. Here, we applied the option in which the covariance matrix is regularized in a fully automatic way, so no hyper-parameter tuning is needed [54,55]. This classifier was selected because it is widely used in many P300-based BCI applications [56] due to its robustness and performance. Technical details of this method can be found in Reference [57].

Additionally, the forward-backward step-wise (SW) method is used to select the characteristics to evaluate in the classification stage. This algorithm integrates the classification model that better contributes to an optimal performance guided by a scoring criterion. By starting with an empty classification model, the best features that improves the performance but are not included in the successive incorporated models are selected. The model remains unchanged if none of the candidate characteristics improves the performance of the classifier. In the next step, that variable in the model that can be excluded without significantly reducing the scoring is eliminated. Once again, if it is not possible to discard a feature without affecting the model, then the feature set remains unchanged. The previous steps are replicated as long as changes to the feature set are possible. The model training and feature selection are performed simultaneously on the used platform.

The combination of CCA spatial filter with regularized LDA classifier shows better performance in the process of single-trial classification of P300 events. Reference [45], as well as Reference [7], studied different options of feature extraction methods combined with classification models, confirming a better performance of this option with respect to other proposals.

### 2.6. Evaluation Process and Metrics

First, the effect of stimulation condition and of the number of symbols on the single-trial classification between target and non-target responses was assessed through a five-fold cross-validation process. Although there exist a sample overlapping, this performance evaluation approach is quite acceptable to estimate the online performance of the BCI [58,59]. Here, all the epochs in a given dataset were randomly allocated into five sets. Four of the sets were used to train the machine-learning model (the CCA-based spatial filter and the LDA classifier), while the remaining set was used to estimate classification accuracy, i.e., the rate of correct classifications for target ($CA_{target}$ or true positive rate), non-target ($CA_{nontarget}$ or true negative rate), and for the total ($CA_{total}=0.5x\left(CA_{target}+CA_{nontarget}\right)$), here we use balanced accuracy due to the imbalance present in the training samples [60]. With this procedure we are avoiding bias toward the non-target class and balance the accuracy calculation. In addition, we calculated the significance levels of the model’s accuracies with a permutation test [61] where the null hypothesis indicates that the observations of both classes are interchangeable, therefore any random permutation of the class labels produces accuracies comparable to those obtained with the non-interchangeable data. The alternative hypothesis is accepted when the accuracy of the model is an extreme value in the empirical distribution constructed with *m* random permutations. When the alternative hypothesis is accepted, we can say that the cross-validation accuracy is above the level of chance. This process was applied independently for each cleaned dataset of each participant and session, and distributions of classification accuracies were then constructed for each stimulation condition, SF and CF, and for each number of symbols, from 4 to 9.

Second, the effects on the classification between target and non-target responses when training and testing the machine-learning model with datasets containing different stimulation conditions or different number of symbols were evaluated as follows. In the case of stimulation condition, the model was trained using the entire cleaned dataset recorded with SF (or CF), while classification accuracy for target, non-target, and the total were computed using the entire cleaned dataset recorded with CF (or SF). Similarly, the case of number of symbols consisted of training the model with an entire cleaned dataset with a given number of symbols and testing performance separately with all the other cleaned datasets with different number of symbols. As an example, if the machine-learning model is trained with the entire cleaned dataset recorded with 4 symbols, then, classification performance is computed separately with each one of the remaining datasets, that is, with the cleaned datasets recorded with 5, 6, 7, 8, and 9 symbols. This was repeated until all number of symbols were used as training set. This procedure was also applied independently for each participant in each session.

We applied the non-parametric Kernel Distribution Estimator (KDE) method [62] to analyze the discrimination process between target and non-target events, for both stimulation conditions. Through this statistical test, significant ERP peaks were identified at each post-stimulus time sample, for each channel, by evaluating with the probability density function (PDF) of the pre-stimulus interval. We established a significance level α, thus identifying as ERP responses all those values that, compared to the pre-stimulus PDF, were greater than 1−α/2 or lower than α/2.

Statistical non-parametric Wilcoxon signed-rank, Wilcoxon rank-sum and Kruskal–Wallis tests were employed to assess significant differences between distributions of classification accuracies for the two visual paradigms and for the six amount of symbols, respectively. All statistical tests were carried out at a confidence level of α=0.05.

## 3. Results

This subsection presents the results of the data analysis procedure which aimed, first, to study the effect of the stimulation conditions and of the number of symbols on the classification accuracy and on the P300 responses, and second, to investigate the effect on the classification accuracy of training and testing the machine-learning model with datasets containing different stimulation conditions and different number of symbols.

### 3.1. Stimulation Conditions

Figure 3 and Figure 4 show the results of the ERP analysis for one participant. These graphs illustrate all the channels for the stimulation conditions, SF and CF, respectively. In each graphic, the signals associated to the target and non-target events are presented in blue and red, respectively. For both stimulation conditions, significant positive and negative components are identified (p<0.05, two-tail test) in a latency of approximately 200 to 600 ms for the waveforms associated with the target events. In contrast, signals associated with non-target events do not manifest significant peaks (p>0.05, two-tail test) in both stimulation conditions. For the SF stimulation condition, all channels, except Fz and Cz, show significant components. In some channels only negative values are generated, such as Pz, P3, and P4, and, in other channels, both polarities are seen, such as PO7, PO8, and Oz. In the CF stimulation condition, all channels manifested significant components with both polarities. It is important to highlight the latency in which these potentials are elicited, which provides strong clues that we are in the presence of ERPs linked to the visual stimuli presented to the participants. The significance graphs for each stimulation condition (Figure 5a,b) show in a complementary format the occurrence by channel of relevant ERP target events for the single-trial classification. It can be noted that the CF condition, with respect to the SF condition, contributes with a greater occurrence of significant events to the process of discriminating target from non-target events, and, at the same time, except for the P3 and PO7 channels, there is specific generation of the P300 component.

The amplitude of the positive peak of the ERP in the target condition was computed for each participant and session. This was carried out for the two types of stimuli (SF and CF). Then, for each electrode, across all participants and sessions distributions of SF and CF were subjected to a statistical analysis. Indeed, significant differences between the medians of the P300 amplitude distributions of SF and CF were found (p<0.05, Wilcoxon rank-sum test) in electrodes P3, Pz, P4, PO7, PO8 and Oz, while, no significant differences were found between the medians of the two distributions (p>0.05, Wilcoxon rank-sum test) for electrodes Fz and Cz.

Table 1 shows the average values of accuracy rate for the two stimulation conditions obtained for each participant across the three sessions. These results show that the stimulus with CF provided the higher accuracy rates in the majority of the participants (17 out of 19). In addition, the across-all-participants averaged accuracy rates for CF and SF were, respectively, 0.824±0.068 (minimum of 0.678±0.033 and maximum of 0.903±0.027) and 0.759±0.069 (minimum of 0.630 and maximum of 0.874), that is, the accuracy rate is 6% greater for CF than for SF. Altogether, considering all participants, stimulation with CF provided greater accuracy rate than SF stimulation condition in the 89.47% of the cases.

Figure 6 shows the across all participants and sessions distribution of accuracy rate for both types of stimuli. Significant differences were found between the two distributions (p<0.05, Wilcoxon rank-sum test) and the median value for CF (0.829) was indeed 5% greater than for SF (0.779). This shows that CF provides significant greater accuracy rates than SF.

Finally, Table 2 shows the accuracy rates for training the classification model with one type of flash and assessing performance with the other type of flash. These results are for all participants in all session and with 5 stimuli only. For comparison purposes, the Table also include the cross-validation results of training and testing with the same type of flash. For the case of training with SF and evaluating with CF, the average of accuracy rate is 0.753, 0.551, and 0.652 for non-target, target, and total, respectively, which are lower accuracies than those obtained in the cross-validation analysis with SF (0.822, 0.803, and 0.812). Similarly, for the case of training with CF and evaluating with SF, the average of accuracy rate is 0.817, 0.412, and 0.619 for non-target, target, and total, respectively. These are also lower accuracies than those obtained in the cross-validation analysis with CF (0.878, 0.848, and 0.863). In the two cases, the median of the distribution of accuracy rate is significantly different (and lower) than the median of the classification accuracy obtained with the cross-validation results (p<0.05, Wilcoxon signed-rank test). This shows that training with one type of stimulus and then testing performance with other type of stimuli reduces performance.

### 3.2. Number of Symbols

The ERP for each number of symbols for the target condition is presented in Figure 7. All ERP show negative peaks between 200 and 300 ms in all electrodes, as well as the characteristic P300 positive peaks between 300 and 400 ms in parietal (P3, Pz, P4), parieto-occipital (PO7, PO8), and occipital (Oz) electrodes. Note that the amplitude of these negative or positive peaks are similar regardless of the number of blinking stimuli.

Table 3 shows the average values of accuracy rate for each number of symbols. These results are for each participant across the three experimental sessions. The accuracy rate is greater in 7, 4, 3, 0, 5, and 0 of the 19 participants for 4, 5, 6, 7, 8, and 9 number of symbols, respectively. The average values across all participants are very similar irrespective of the number of symbols (see the values presented at the bottom of the Table), indeed, the minimum and maximum accuracy rate are 0.790 and 0.816, respectively, where there is only a difference of 0.026.

To examine significant differences, Figure 8 shows the across all participants and sessions distribution of accuracy rate for each number of symbols. No significant differences were found in the distributions of accuracy rate for the different number of symbols (p=0.628, Kruskal–Wallis test), which indicates the same classification accuracy regardless of the number of blinking stimuli. We also examined significant differences of the classification accuracy across all number of symbols separately for the two stimulation conditions (this is possible since the experiments of the number of symbols were carried out with the stimulation condition that provided the greater accuracy rate, that is 89.47% of the cases with CF and the rest of the cases with SF). In both stimulation conditions, no significant differences were found in the accuracy rate across the different number of symbols (p>0.05, Kruskal–Wallis test). These results suggest the same classification accuracy regardless of the number of symbols irrespective of the stimulation condition employed in the study.

Finally, Table 4 shows the accuracy rate results for training the machine learning model with a dataset recorded with a given number of symbols and assessing performance separately with each one of the remaining datasets recorded with a different number of symbols.

For comparison purposes, the cross-validation results of training and testing with datasets containing the same number of symbols are also included (gray-highlighted values in the diagonal). These results shows that the accuracy rates are very similar irrespective of the number of symbols employed to train and test the model and that they are also similar to the cross-validation results of training and testing with the same number of symbols. For instance, for the case of training with datasets with 5 symbols, the minimum and maximum accuracy rates are 0.798, 0.699, and 0.748 for datasets with 9 stimulus, and 0.839, 0.696, and 0.768 datasets with 4 symbols (for non-target, target, and total, respectively), while the cross-validation results are 0.841, 0.771, and 0.806. For each case, no significant differences were found in the distributions of accuracy rate across the different number of symbols (p>0.05, Kruskal–Wallis test). This shows that training with a dataset with a given number of symbols and then testing performance with other datasets containing different number of symbols results in similar performance.

### 3.3. Participant and Session

To examine the effect on the performance across participants and sessions, Figure 9 shows the distribution of accuracy rates across-all number of symbols separately for each participant and for each session. For the case of participants (Figure 9a), significant differences were found between the median of distributions of accuracy rate (p<0.05, Kruskal–Wallis test), with high variability in the median values ranging from 0.673 for participant 2 up to 0.899 for participant 18. For the case of sessions (Figure 9b), significant differences were also found between the median of distributions of accuracy rate (p<0.05, Kruskal–Wallis test) with median values of 0.782, 0.808, and 0.831 for sessions 1, 2, and 3, respectively, which indicates that the performance increases as more sessions are carried out.

## 4. Discussion

Some aspects of the visual stimuli (e.g., shape, color, type of stimulation, number of options, among others) in P300-based BCIs may affect the characteristics of the ERP; thus, they play a critical role in the BCI performance. Some previous works have explored these issues; however, further investigation is still required to gain more understanding of the effect that such parameters have on the ERP and on the BCI performance. The first goal in this research was to compare the effects of the stimulation conditions standard flash (SF) and cartoon face (CF), and of the number of symbols that are intensified (from four to nine), on the ERP responses and on the classification accuracy between targets and non-target events. The second goal was to assess the influence of training the machine learning model that recognizes between targets and non-target events with a dataset recorded with a stimulation condition or a number of symbols and assess classification accuracy with other datasets containing a different stimulation condition or number of symbols. An additional aim of this work was to provide a dataset of P300 EEG recordings to study and evaluate signal processing and machine-learning algorithms for P300-BCIs considering two stimulation conditions, several number of symbols, and several sessions.

Considering the stimulation condition, the analysis of the ERP showed higher amplitude values with CF than with SF, which were more notorious in channels PO7, PO8, and Oz in the 200–300 ms interval (see Figure 3 and Figure 4 as an example). This indicates that stimulation condition based on CF produces stronger ERP responses than the classical SF. In addition, this result about the posterior location of ERP significant responses is consistent with results in Reference [47] and so is the fact that responses associated with non-target events exhibit a sinusoidal pattern of visual evoked steady-state potential that coincides with the frequency of stimulation. On the other hand, the significantly greater classification accuracy achieved with CF (Figure 6), along with the almost 90% of the cases where CF provided the greatest performance among all participants and sessions (Table 1), also indicated that such stimulation condition provided better classification accuracy. A similar previous work also suggested that CF provides better accuracy [37], nonetheless, the stimulation in that work includes facial expression changes. Though this is different to the CF stimulation employed in our experiments, it reinforces that CF is better than SF. We should point out that we did not study the relevance of facial expression or emotional states in the CF, however, we must highlight that using “happy” expressions agrees with previous works suggesting the use of positive expressions [63,64]. In summary, these results indicated that CF as stimulation strategy should be preferred over the classical SF for P300-BCIs in specific applications of target selection. However, there is always the possibility that some participants feel more comfortable with standard flashes.

Regarding the number of symbols, the analysis of the ERP showed the same waveforms, peak amplitudes and latencies irrespective of the number of stimuli (see Figure 7). Hence, there are no differences in the ERPs associated with the number of blinking options presented to the user. Likewise, no significant differences in the classification accuracy were found among the number of symbols (see Figure 8). Indeed, the maximum and minimum accuracy (averaged across-all-participants) among all number of symbols presented a slightly difference of only 2.6% (see Table 3). A previous work explored 3 configurations of rows-columns visual paradigm where the total number of options that were intensified varied (4 × 4, 8 × 8 and 12 × 12 matrix), and despite the differences with our single-option visual paradigm, they also found that different number of symbols generates ERPs with greater amplitude in the posterior locations (parietal and parietal-occipital areas, respectively) of both hemispheres [40]. On the other hand, we found no evidence suggesting differences in the amplitudes of the ERPs and accuracy when applying different numbers of stimuli. The results presented in this analysis indicate that the amount of stimulation symbols shown to the participants does not vary the ERP signatures neither the accuracy in the recognition between target and non-target responses.

Additional analyses showed that the overall classification accuracy increase in most of the participants as new sessions were carried out. This aspect is noteworthy as it possibly suggests an adaptation of the users to the experimental environment and would be appropriate to consider for studies where more than one session is required. In contrast, the variability in performance across participants is high, which indicates that there is no relationship in the inter-subject performance. This supports the individual nature of the BCI experience for each participant, particularly for those based on P300 control signals.

Another critical aspect studied herein was the performance of the machine learning model when training and testing using datasets recorded with different conditions of stimulation and different number of symbols. This is important to ascertain whether the calibration and the online usage of P300-BCIs can employ different stimuli parameters, which can occur when switching the P300-BCI application due to, for example, changes in the disease state of patient users or improvements in robotic devices. On the one hand, training with one stimulation type and testing with the other type leads to a reduction of classification accuracy (see Table 2). This reduction is larger for target than for non-target events, which is expected since we are just changing the visual scheme that evokes target responses. Importantly, the accuracy of target events is also more reduced when training with CF and testing with SF than vice-versa. This behavior is consistent since CF stimulation elicits stronger potentials and this makes it easier for the model to identify target events. However, when testing with signals containing lower amplitude resulting from SF stimulation, then the discrimination process would be affected.

Training with a given number of symbols and testing with other number of symbols showed no changes in classification accuracy (see Table 4). Therefore, the variation of the number of symbols presented during calibration and online operation does not imply changes in the recognition between target and non-target responses. These results have important implications for P300-BCIs (at least in single-option stimuli paradigm configurations) because the calibration stage and the online usage of the system: (i) should be carried with visual interfaces with the same stimulation condition to avoid reduction in performance; (ii) can be carried out with visual interfaces with different number of symbols without affecting performance, even, the calibration stage can be carried out with a low number of symbols to decrease the calibration time and thus reduce fatigue and boredom to the users.

Previous works have reported single-trial classification rates in the order of 0.8 (see Reference [58,59,65]). The classification rates reported are similar to those achieved in this work. It is important to mention that the single-trial classification results reported herein are meaningful for BCI systems since the accuracy in the recognition of the option the user is attending to in online settings is boosted by classifying multiple instances to address the target and non-target selection.

In a different front, we want to point out that no questionnaire or survey was conducted to qualitatively determine the level of comfort and/or the user’s preference with the two visual stimulation conditions studied herein. However, at the end of each experimental session, each participant was verbally asked about his/her predilection, and there was a generalized preference for CF as stimulation type, while the preference for the number of symbols varied greatly from subject to subject.

To sum up, the present work showed that: (i) the stimulation with cartoon faces is superior to the stimulation with standard flash since it generates ERPs with larger amplitudes and favors the appearance of other components that contribute to a more discriminative power of the target events with respect to the non-target ones; (ii) stimulation with different number of symbols offers no difference in performance so it is appropriate to perform the calibration process with as few stimuli as possible to decrease the training time; (iii) the single-trial classification between target and non-target events improves as users become more familiar with the interface and its environment; and (iv) target and non-target events can be discriminated with an appropriate level of confidence in datasets obtained by varying properties of the visual stimulus coming from the same subject and during the same session.

Finally, the recorded EEG signals are freely available and they can be used for the study and evaluation of signal processing and machine-learning models in P300-BCI (see Appendix A). The future work will explore new classification algorithms that would be worth incorporating [66], also include the understanding of the implication of changing the parameters of the stimuli (scheme and number) between offline and online P300-BCI operation, and the inter-session and inter-subject performance evaluation tests both in offline and online settings.

## Figures and Tables

**Figure 1 sensors-20-07198-f001:**
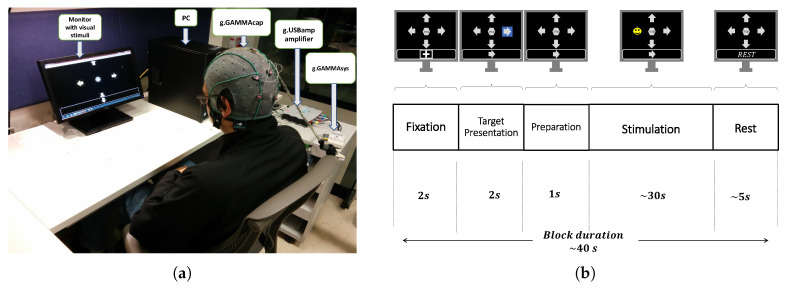
Description of the experimental paradigm. (**a**) Picture of the experimental setup with a participant, the computer screen with the Graphical User Interface (GUI) displaying a set of 5 stimuli and the instruction box, and the electroencephalogram (EEG) recording system. (**b**) Illustration of the temporal sequence of a block. Each block consists of five phases: Fixation, Target Presentation, Preparation, Stimulation, and Rest.

**Figure 2 sensors-20-07198-f002:**
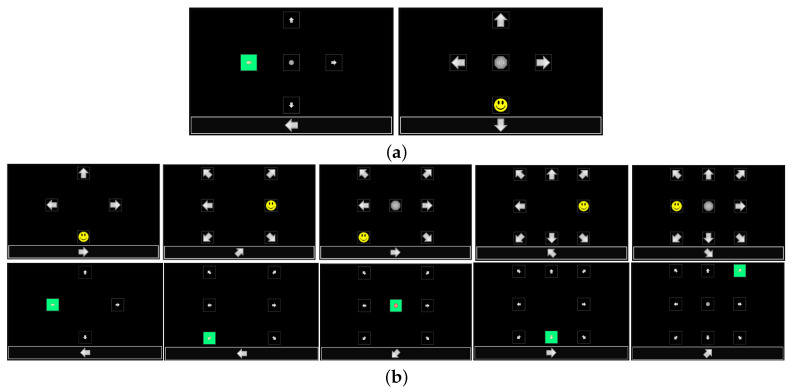
Screenshots of the GUI with the two visual configurations under study. (**a**) Illustration of the two stimulation conditions with the configuration for 5 symbols. Left panel: standard flash based on green-highlight of the stimulus or stimulus (SF). Right panel: superimposing a yellow smiling cartoon face or cartoon face (CF). (**b**) Illustration of the configuration on the screen for 4, 6, 7, 8, and 9 symbols for both stimulation conditions. Note that, in all cases, the symbols are evenly distributed on the screen and the information box is in the bottom.

**Figure 3 sensors-20-07198-f003:**
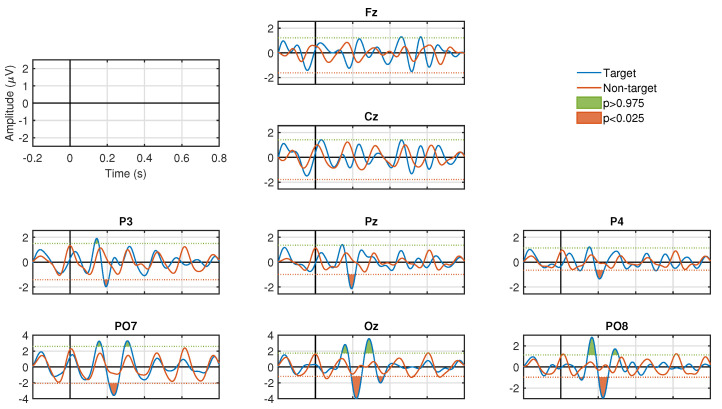
ERP responses for all channels in one participant for the target (blue signal) and non-target (red signal) events used in single-trial classification for SF stimulus. Reported signal-to-noise ratios (target vs. non-target): Fz-3.63 dB, Cz-0.48 dB, P3-0.88 dB, Pz-3.20 dB, P4-2.99 dB, PO7-1.86 dB, PO8-4.02 dB, Oz-4.26 dB. Green and orange areas in the ERP correspond to the positive and negative peaks that presented significant differences (p<0.05, two tail test) with the estimated Probability Density Function (PDF) of the baseline period. No significant peaks are observed in the ERP for the non-target condition.

**Figure 4 sensors-20-07198-f004:**
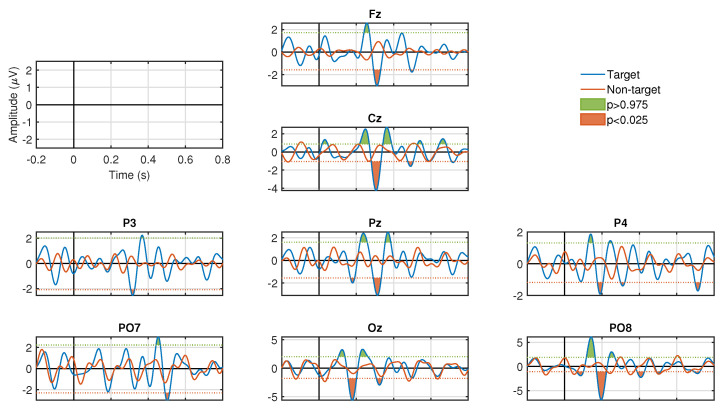
ERP responses for all channels in one participant for the target (blue signal) and non-target (red signal) events used in single-trial classification for CF stimulus. Reported signal-to-noise ratios (target vs. non-target): Fz-10.38 dB, Cz-7.10 dB, P3-8.46 dB, Pz-7.51 dB, P4-6.36 dB, PO7-4.89 dB, PO8-6.05 dB, Oz-4.46 dB. Green and orange areas in the ERP correspond to the positive and negative peaks that presented significant differences (p<0.05, two tail test) with the estimated PDF of the baseline period. No significant peaks are observed in the ERP for the non-target condition.

**Figure 5 sensors-20-07198-f005:**
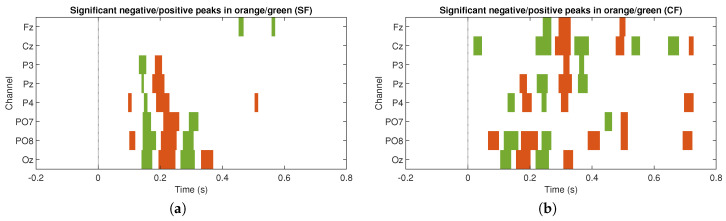
Statistical significance for all channels in one participant for the target and non-target events used in single-trial classification for (**a**) SF stimulus and (**b**) CF stimulus.

**Figure 6 sensors-20-07198-f006:**
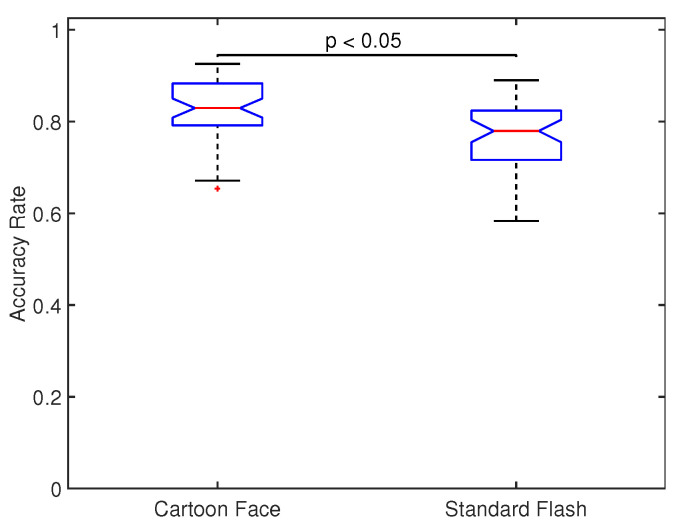
Across all participants and sessions, distribution of accuracy rates for both types of stimuli (CF and SF). Significant differences were found between the two distributions (p<0.05, Wilcoxon rank-sum test) with median values of 0.829 and 0.779 for CF and SF, respectively.

**Figure 7 sensors-20-07198-f007:**
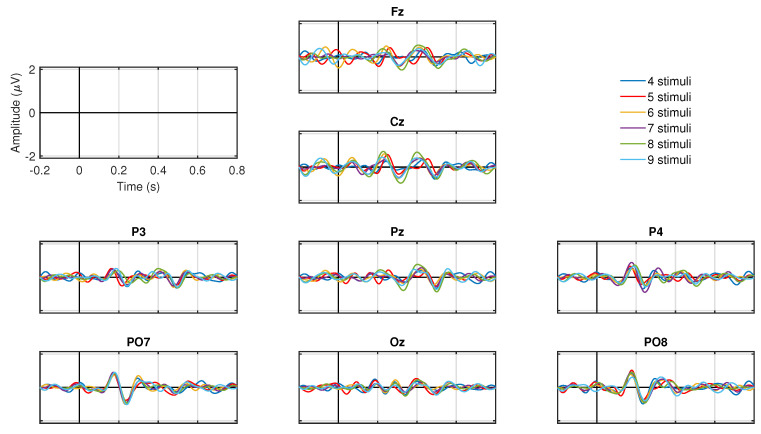
Across all participants and sessions EPR target responses for the different number of symbols (from four to nine).

**Figure 8 sensors-20-07198-f008:**
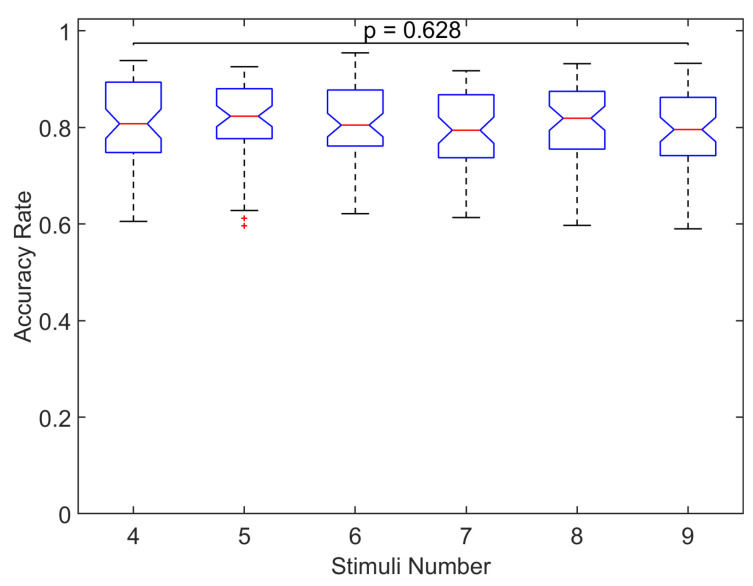
Across all participants and sessions distribution of accuracy rates for each number of symbols. No significant differences are found between the median of distributions (p=0.628, Kruskal–Wallis test).

**Figure 9 sensors-20-07198-f009:**
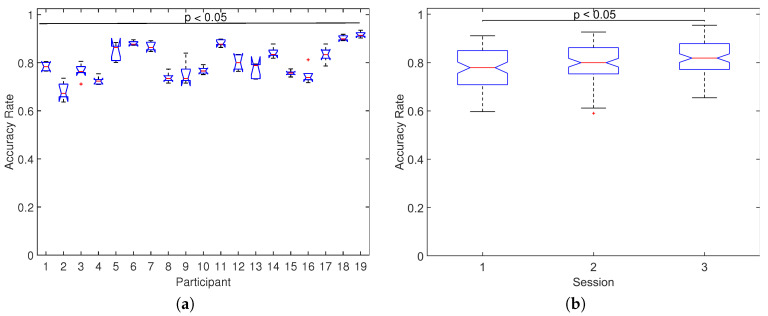
Across all participants and sessions distribution of accuracy rates obtained for (**a**) each participant in all sessions, significant differences are verified (p<0.05, Kruskal–Wallis test), and (**b**) each session in all participants, significant differences are verified (p<0.05, Kruskal–Wallis test). These results are for all number of stimulus.

**Table 1 sensors-20-07198-t001:** Average value of accuracy rate for the two stimulation conditions obtained for each participant in all sessions. The value in bold for each participant is the highest between both types of stimuli.

Participant	Stimuli Type	
	Cartoon Face	Flash
**1**	0.823±0.004	0.822±0.026
**2**	0.678±0.033	0.630±0.000
**3**	0.813±0.101	0.719±0.000
**4**	0.797±0.002	0.723±0.030
**5**	0.878±0.006	0.734±0.028
**6**	0.898±0.016	0.874±0.022
**7**	0.885±0.038	0.808±0.056
**8**	0.774±0.036	0.621±0.044
**9**	0.847±0.034	0.805±0.062
**10**	0.792±0.041	0.671±0.069
**11**	0.876±0.011	0.794±0.028
**12**	0.833±0.012	0.787±0.010
**13**	0.781±0.024	0.736±0.076
**14**	0.878±0.051	0.835±0.019
**15**	0.728±0.052	0.743±0.077
**16**	0.709±0.074	0.717±0.082
**17**	0.878±0.051	0.835±0.019
**18**	0.895±0.012	0.765±0.045
**19**	0.903±0.027	0.800±0.039
**Average**	0.824±0.068	0.759±0.069

**Table 2 sensors-20-07198-t002:** Across-all participants and sessions accuracy rates for training the machine learning model with one type of flash and assessing performance with the other type of flash. The cross-validation accuracy rate results of training and testing with the same type of flash are also included (gray-highlighted values).

	Training Stimulus Type					
**Test Stimuli Type**	**SF**			**CF**		
	**Non-Target**	**Target**	**Average**	**Non-Target**	**Target**	**Average**
**SF**	0.822±0.059	0.803±0.056	0.812±0.057	0.817±0.060	0.412±0.107	0.619±0.060
**CF**	0.753±0.058	0.551±0.099	0.652±0.066	0.878±0.060	0.848±0.056	0.863±0.057

**Table 3 sensors-20-07198-t003:** Average value of accuracy rate for each number of symbols obtained for each participant in all sessions. The value in bold for each participant is the highest accuracy rate among all number of symbols.

Participant	Number of Symbols					
	4	5	6	7	8	9
**1**	0.804±0.064	0.765±0.098	0.787±0.082	0.801±0.087	0.767±0.102	0.780±0.075
**2**	0.711±0.003	0.665±0.033	0.735±0.018	0.680±0.026	0.658±0.060	0.637±0.041
**3**	0.759±0.116	0.711±0.075	0.784±0.123	0.762±0.121	0.805±0.114	0.764±0.111
**4**	0.712±0.096	0.730±0.115	0.710±0.077	0.728±0.100	0.754±0.072	0.719±0.077
**5**	0.885±0.028	0.865±0.024	0.869±0.016	0.809±0.037	0.862±0.025	0.801±0.019
**6**	0.873±0.048	0.875±0.042	0.879±0.015	0.875±0.043	0.896±0.023	0.887±0.027
**7**	0.891±0.021	0.885±0.038	0.861±0.101	0.847±0.045	0.866±0.062	0.854±0.057
**8**	0.746±0.017	0.774±0.036	0.729±0.019	0.715±0.031	0.726±0.022	0.740±0.022
**9**	0.840±0.061	0.774±0.012	0.729±0.032	0.715±0.050	0.726±0.061	0.740±0.045
**10**	0.772±0.039	0.792±0.041	0.776±0.014	0.751±0.036	0.757±0.040	0.759±0.036
**11**	0.898±0.011	0.876±0.011	0.878±0.023	0.864±0.023	0.877±0.010	0.895±0.007
**12**	0.810±0.048	0.833±0.012	0.835±0.029	0.764±0.022	0.791±0.030	0.773±0.034
**13**	0.734±0.044	0.790±0.009	0.787±0.020	0.792±0.024	0.795±0.032	0.732±0.040
**14**	0.837±0.078	0.878±0.051	0.830±0.075	0.819±0.091	0.852±0.051	0.832±0.027
**15**	0.741±0.063	0.751±0.071	0.762±0.030	0.756±0.007	0.774±0.060	0.758±0.012
**16**	0.812±0.102	0.717±0.063	0.738±0.075	0.729±0.085	0.739±0.081	0.755±0.057
**17**	0.837±0.078	0.878±0.051	0.830±0.075	0.819±0.091	0.852±0.051	0.786±0.027
**18**	0.912±0.004	0.895±0.012	0.918±0.013	0.900±0.010	0.898±0.006	0.891±0.011
**19**	0.935±0.005	0.903±0.027	0.910±0.005	0.916±0.002	0.925±0.002	0.912±0.029
**Average**	0.816±0.084	0.808±0.085	0.808±0.077	0.792±0.080	0.806±0.082	0.790±0.080

**Table 4 sensors-20-07198-t004:** Across-all participants and sessions accuracy rates for training the classification model with a dataset recorded with a given number of symbols and assessing performance separately with other datasets recorded with different number of symbols. The cross-validation accuracy rate results of training and testing with the dataset with the same number of symbols are also included (gray-highlighted values in the diagonal).

		Number of Symbols in the Training Dataset					
Testing		*4*	*5*	*6*	*7*	*8*	*9*
***4***	**Non-Target**	0.841±0.085	0.839±0.083	0.854±0.083	0.858±0.078	0.864±0.077	0.859±0.076
	**Target**	0.779±0.102	0.696±0.119	0.701±0.110	0.686±0.106	0.670±0.110	0.637±0.107
	**Average**	0.812±0.090	0.768±0.096	0.778±0.093	0.772±0.089	0.767±0.089	0.748±0.086
***5***	**Non-Target**	0.796±0.080	0.841±0.079	0.834±0.079	0.840±0.074	0.846±0.081	0.846±0.072
	**Target**	0.737±0.109	0.771±0.102	0.687±0.110	0.689±0.101	0.658±0.112	0.652±0.108
	**Average**	0.767±0.090	0.806±0.090	0.761±0.087	0.765±0.083	0.751±0.087	0.749±0.084
***6***	**Non-Target**	0.792±0.081	0.813±0.080	0.863±0.065	0.840±0.072	0.852±0.072	0.846±0.070
	**Target**	0.762±0.110	0.708±0.140	0.829±0.059	0.737±0.120	0.729±0.119	0.704±0.113
	**Average**	0.779±0.088	0.761±0.099	0.846±0.061	0.789±0.088	0.791±0.086	0.775±0.082
***7***	**Non-Target**	0.782±0.076	0.807±0.073	0.823±0.075	0.855±0.066	0.841±0.069	0.842±0.068
	**Target**	0.740±0.105	0.708±0.109	0.732±0.094	0.816±0.058	0.725±0.087	0.726±0.093
	**Average**	0.761±0.085	0.757±0.085	0.778±0.078	0.835±0.060	0.782±0.074	0.783±0.078
***8***	**Non-Target**	0.776±0.080	0.803±0.074	0.821±0.080	0.827±0.075	0.861±0.067	0.835±0.074
	**Target**	0.753±0.111	0.717±0.122	0.752±0.103	0.753±0.101	0.822±0.066	0.736±0.099
	**Average**	0.764±0.090	0.760±0.091	0.787±0.088	0.790±0.084	0.841±0.065	0.785±0.082
***9***	**Non-Target**	0.770±0.075	0.798±0.074	0.815±0.077	0.827±0.074	0.835±0.070	0.854±0.066
	**Target**	0.727±0.108	0.699±0.111	0.721±0.107	0.740±0.095	0.729±0.096	0.811±0.061
	**Average**	0.748±0.084	0.748±0.084	0.768±0.083	0.783±0.078	0.782±0.077	0.838±0.064

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
