# Peer review of "Single-Option P300-BCI Performance Is Affected by Visual Stimulation Conditions"

_sensors, 2020, doi:10.3390/s20247198_

Round 1

Reviewer 1 Report

This paper analyzes the effect of a smiley face highlight instead of flashing on the P300 and it's respective classification.

Overall, this paper misses substantial information for assessment and seems not yet finished.

  1. The structure of the paper is not adhering to scientific standards. It also does not fit to the description in the introduction. The results are described before the method and data description. The conclusion is completely missing.
  2. The state of the art is rather short. In the field of P300 detection with ML methods, the work by the Berlin BCI group lead by Klaus-Robert Müller needs to be discussed. For single trial, P300 detection, the work by Elsa Andrea Kirchner might be relevant, too. For “Indeed, parameters such as shape, size, color and type of flash may enhance or diminish the difference between target and non-target responses, and thus, may influence the performance of the machine-learning model.” a reference is missing. But given this is supposed to be a well know fact, the purpose of the paper is not really clear.
  3. The relation of the paper of the state of the art is not fully worked out. The authors do not really point out what their contribution is and even less why it really matters. They mention work with stimuli using face images. The author's experiments do not include this kind of stimulus for a fair comparison and do not sufficiently compare to the respective experimental design and results. Hence, this publication seems rather incremental and does not sufficiently compare to the state of the art.
  4. What does “Single-option” in the title mean? It seems confusing and is not explained.
  5. The experimental design is not sufficiently described. What was the overall timing of the experiments? How many days passed between the three recording sessions? How were the resistance values of the cap after the sessions? One could guess that the experiments were conducted exactly in the order, listed in the paper. This is a major experimental flaw. There are several effects related to P300 that change over time. At the beginning, the subject gets used to the experiments and might have more wrong P300s from non-targets and higher amplitudes due to lack of familiarity. Due to fatigue, amplitudes might decrease and wrong reactions to non-targets might occur. To the best of my knowledge, randomized order of experiments is required in this case. Why were only that few electrodes recorded?
  6. The changes concerning the different number of symbols could be also caused by a longer avg. time between flashes and be actually independent of the number of symbols.
  7. The discussion section needs more structure.
  8. The design of the experiment seems to deviate from other experiments. Hence, a comparison is not possible.
  9. Since samples overlap in the experiments, using randomized 5-fold cross-validation might cause contamination of training and test data. In the classification evaluation, it is also not clear, how the regularization/shrinkage was tuned. For P300, hyperparameter tuning can be very influential. To the best of my knowledge, 3 channels from CCA are not sufficient. They might even remove certain artifacts but not not emphasize the signal of interest. The description of the performance metric needs improvement. Accuracy and Positiva/negative classification rates have different meanings. Standard metrics like Balanced Accuracy should be used.
  10. The curves in Figure 1 are too small. Using red and green is potentially suboptimal for color blind people. The curves from the experiments look quite noisy. This should be explained.
  11. There are numerous grammar errors:
    • “many research”
    • line 67: “Despite that several works has demonstrated”
    • line 142: “the two stimulus type”
    • line 346: This last 1s and indicates to be ready for the upcoming phase.
    • line 352: This phase last around 30s which slightly varies according to the number of symbols.
    • line 353: None of the symbol …
    • “a custom-made software (SW) that manage”
    • “previous experience in the use of BCI”

In summary, this paper seems to need substantial rework to be useful for the scientific community.

Author Response

1. The structure of the paper is not adhering to scientific standards. It also does not fit to the description in the introduction. The results are described before the method and data description. The conclusion is completely missing.

Response: Thank you very much for your comments. We apologize for the unconventional structure of our document and its effect on the review of our proposal. This structure fully adheres to the format that appears in the MDPI (Sensors)’s author guidelines: https://www.mdpi.com/journal/sensors/instructions They suggest to use the corresponding template in Overleaf for authors of articles (https://www.mdpi.com/authors/latex) in section “Software and Apps for Writing LaTeX,” subsection “Online with Overleaf.” We also found this structure odd. However, we assumed that it had this order to make the review process more expeditious and so we did not change it. However, the document has already been structured according to the following order: Introduction, Materials and Methods, Results, and Discussion. We consider that the current structure of the proposal contributes to a better interpretation.

Similarly, the study conclusions are concretely stated from line 503 in the “Discussion” section of the new version of the manuscript.

An explicit “Conclusions” section was not included following the author’s guidelines (https://www.mdpi.com/journal/sensors/instructions), Section “Manuscript Preparation,” subsection “Research Manuscript Sections” and also in the latex template. About this section is outlined in both places: “This section is not mandatory, but can be added to the manuscript if the discussion is unusually long or complex.”

2. The state of the art is rather short. In the field of P300 detection with ML methods, the work by the Berlin BCI group lead by Klaus-Robert Müller needs to be discussed. For single trial, P300 detection, the work by Elsa Andrea Kirchner might be relevant, too. For “Indeed, parameters such as shape, size, color and type of flash may enhance or diminish the difference between target and non-target responses, and thus, may influence the performance of the machine-learning model.” a reference is missing. But given this is supposed to be a well know fact, the purpose of the paper is not really clear.

Response: Thank you for your comments. We have incorporated some valuable contributions by Elsa Andrea Kirchner and the Berlin BCI group lead by Klaus-Robert Müller in “1. Introduction” and “2.5.2. Feature extraction” sections, respectively. For distinction, the added text has been highlighted in red.

For our study, we consider that it is essential to include references related to visual stimulation paradigms and their effect on BCI performance. This aspect is covered in paragraphs 3, 4, and 5 of the introductory section, where we analyze papers that study different stimulation methods for P300-based BCIs. Additionally, we consider previous works that compare visual stimulation with faces against the conventional flash stimulation. Finally, we include works that proposed the use of cartoon faces to replace the stimulation with faces.

It is important to say that the influence on performance from varying the number of symbols shown on the screen has not been studied previously for single-option P300-based BCIs, so we consider that the study presented in our work is relevant to the BCI community.

Considering your observation regarding the purpose of our study, we have re-elaborated the first sentence of paragraph 6 (highlighted in red) in section “1. Introduction” to clarify and make more explicit the purpose of our study.

3. The relation of the paper of the state of the art is not fully worked out. The authors do not really point out what their contribution is and even less why it really matters. They mention work with stimuli using face images. The author’s experiments do not include this kind of stimulus for a fair comparison and do not sufficiently compare to the respective experimental design and results. Hence, this publication seems rather incremental and does not sufficiently compare to the state of the art.

Response: We hope that the previous explanation about the structure of the state of the art has contributed to a better understanding of the reason for the related works in that section.

We have considered your feedback and have corrected the text, explicitly indicating our contribution in paragraph 6 of the introductory section (highlighted in red), which is: “our contribution is addressed to the optimization of BCI performance from establishing design parameters in single-option visual stimulation paradigm, where only one element of those presented in the visual scheme, flashes at a time and whose application can be found in navigation systems or remote control of devices, in addition to the clinical environment.” The significance of our proposal is that “sets standards in P300 BCI’s visual interface design for target selection applications. The results also showed that the classification performance of training and testing the machine learning model is reduced when using datasets containing a different type of stimulus but is similar regardless of the number of symbols. This is important because it would yield answers about the optimal number of symbols required to train the interface”.

We did not consider human faces in our study because, according to references 34, 35, and 36, the stimulation with cartoon faces is not significantly different from the stimulation with human faces regarding the amplitudes of the generated ERPs and the BCI performance. Besides that, there are no copyright limitations for using cartoon faces.

4. What does “Single-option” in the title mean? It seems confusing and is not explained.

Response: “Single-option” is a visual stimulation paradigm in which only one element of the visual scheme flashes at a time. We wanted to separate it from other configurations such as the classic row-column configuration, checkerboard, Rapid Serial Visual Presentation (RSVP) structure, or other existing options. The “Single-option” stimulation method is useful for applications in which the number of options is not high, for instance, navigation controls. In some references, can be associated with the terms “single-character” or “single-item.” [Jin (2014), Chen (2015), Chen (2016)]

From this observation, we have defined this term more clearly in the sixth paragraph of the introductory section (text highlighted in red).

5. The experimental design is not sufficiently described. What was the overall timing of the experiments? How many days passed between the three recording sessions? How were the resistance values of the cap after the sessions? One could guess that the experiments were conducted exactly in the order, listed in the paper. This is a major experimental flaw. There are several effects related to P300 that change over time. At the beginning, the subject gets used to the experiments and might have more wrong P300s from non-targets and higher amplitudes due to lack of familiarity. Due to fatigue, amplitudes might decrease and wrong reactions to non-targets might occur. To the best of my knowledge, randomized order of experiments is required in this case. Why were only that few electrodes recorded?

Response: The overall time per session was 50 minutes and was already incorporated into the amended document from line 142 of section 2.1 “Experimental protocol” and also, the number of days passed between the three recording sessions was 21 days maximum and 10 days minimum (line 172, text highlighted in red).

We assume that when the reviewer refers to the “resistance values of the cap,” it inquires about the electrodes’ impedance. In this case, those values are reported in the document (line 183). We verified the impedances and contact qualities of the electrodes during the experimental sessions according to the recommended values for active electrodes. We also visually checked the quality of the EEG signals in our experiments, and before starting each experiment, we instructed the users to avoid blinking, jaw movements, and other possible sources of noise and artifacts.

As explained in the experimental protocol description (line 164), we run the experiment block for the two visual stimulation conditions (standard flash and cartoon face) in random order for each participant. Likewise, the number of flashing symbols was varied randomly. This procedure is also described in the document (line 169).

Finally, the electrode configuration selected for our study is widely employed for P300-based BCIs, which is evidenced in references 45, 46 and 47.

6. The changes concerning the different number of symbols could be also caused by a longer avg. time between flashes and be actually independent of the number of symbols.

Response: Our study revealed that the machine learning model performance is not significantly affected by the number of symbols. This statement is one of the contributions of our work.

7. The discussion section needs more structure.

Response: From your observation, we made some changes in the Discussion section to improve its readability. Now, the discussion section is structured as follows:

Paragraph 1 describes the importance of studying the visual stimulus’s properties for developing practical BCIs. Also, it describes the work objectives again.

Paragraphs 2 discusses the results according to the first objective, which is the analysis of the effect of the stimulus condition on the classification performance.

Paragraphs 3 discusses the results according to the second objective, which is the analysis of the effect of the number of symbols on the classification performance.

Paragraph 4 discusses other study results focusing on the analysis by participants and by session.

Paragraphs 5 and 6 describe two other results obtained in our study that evaluate the system’s performance when trained with data from different subjects, sessions, and stimulation conditions. These paragraphs also indicate the relevance of this analysis for different BCI applications.

Paragraph 7 quantitatively compares the accuracy values obtained in our study with the performance described in other works.

In paragraph 8, we identified one of the limitations of our study.

Paragraphs 9 and 10 report the conclusions of our proposal.

8. The design of the experiment seems to deviate from other experiments. Hence, a comparison is not possible.

Response: The experiment design was inspired by previous works such as Jin(2014) and Kapgate (2019). Both studies reported the system performance for different visual stimulation conditions to determine possible optimal configurations. We followed the same idea to design our experiment, but our study has other objectives. In our case, we believe that it is essential (and novel) to evaluate the performance of the machine learning block when the number of symbols and the visual stimulation conditions are varied. In the analysis of our results, we compared studies that, although they pursue other objectives, show similarities with our proposal in terms of performance metrics [Krusienski (2008), Jin (2014)].

9. Since samples overlap in the experiments, using randomized 5-fold cross-validation might cause contamination of training and test data. In the classification evaluation, it is also not clear, how the regularization/shrinkage was tuned. For P300, hyperparameter tuning can be very influential. To the best of my knowledge, 3 channels from CCA are not sufficient. They might even remove certain artifacts but not not emphasize the signal of interest. The description of the performance metric needs improvement. Accuracy and Positiva/negative classification rates have different meanings. Standard metrics like Balanced Accuracy should be used.

Response: Thank you for your comment. We are aware of this problem because it is has been observed in many P300-based BCIs. We know that two target trials may overlap in time because sometimes the distance between them is about 300 ms. At the same time, the distance between non-target trials may be 150 ms. We could reduce this overlapping by increasing the time between flashes, but in our experience, increasing the time between trials does not improve the system performance and does not produce higher evoked responses.

As a consequence of this overlapping, the k-fold cross-validation results may be overestimated. However, the error estimation obtained with k-fold cross-validation is a good indicator of the expected performance in online conditions. For example, in Delijorge (2020) and Mendoza-Montoya (2018) (both cited in our paper), it is possible to observe that higher cross-validated accuracies (calculated with k-folds) lead to higher information transfer rates (ITR). The same effect can be observed in other works where the same cross-validation scheme was used (Tanaka(2019), Wirth (2020)). In our study, both stimulation paradigms (cartoon face and standard flash) are evaluated under the same conditions. In this way, the comparison is valid in our study, and the same conclusion would be obtained with another cross-validation scheme.

From your observation, we have added in the document the limitation of using k-fold cross-validation in section “2.6 Evaluation process and metrics” (text highlighted in red). As stated above, this performance metric is acceptable to estimate the online performance of the BCI and can be used to compare experimental conditions. Additionally, we provided a comparison between ERPs, which help to bring support to our conclusion.

On the other hand, we have enhanced the document with a more detailed explanation in section “2.5.3 Classifier” (text highlighted in red). In our system, the covariance matrix is calculated in a fully automatic way, using the method proposed by Ledoit[2004] and described in Lotte[2009], where the shrinkage parameter is obtained directly from the sample covariance. In this way, we do not need to estimate any hyperparameter in the training phase.

We agree with the reviewer that CCA improves the Signal-to-Noise Ratio when it is used as a spatial filtering technique. We have tested this technique with other databases, and for the electrode arrangement selected for our study, we do not see any improvement when we increase the number of spatial filters. Additionally, in the feature selection technique, the subset of relevant variables usually corresponds to the first three spatial filters. We can increase this parameter, but we would obtain similar performance. Additionally, as this is an important aspect, we have incorporated in the section (“2.5.2. Feature extraction”) the reference of Spuler(2014) where it is stated that the use of all the calculated spatial filters is not mandatory (highlighted in red).

Finally, we appreciate your observation regarding the imbalanced sample. The accuracy for each class (target and non-target) is reported in our document, and the system performance is calculated as the average of both quantities. In this way, we are using the weighted accuracy to evaluate the classification rate. To improve the understanding of this aspect, we have added a more detailed explanation of the determination of this parameter in section “2.6 Evaluation process and metrics” (text highlighted in red).

10. The curves in Figure 1 are too small. Using red and green is potentially suboptimal for color blind people. The curves from the experiments look quite noisy. This should be explained.

Response: Thank you for your comments. We split the figures to increase their size and improve their interpretability.

Based on your observations, we investigated the appropriate colors for colorblind people. We found that, coincidentally, the orange and green colors used in our plots are included in one of the primary palettes recommended by Masataka Okabe and Kei Ito (referred in NATURE, Vol. 445, 8, February 2007, https://doi.org/10.1038/445593c).

Finally, the explanation of the oscillatory- rather than noisy- characteristic of the signals responds to the manifestation of steady-state visually evoked potentials produced by the visual stimulus shown to the participants. This phenomenon is explained in section “4. Discussion”, paragraph 2.

11. There are numerous grammar errors:
o “many research”
o line 67: “Despite that several works has demonstrated”
o line 142: “the two stimulus type”
o line 346: This last 1s and indicates to be ready for the upcoming phase.
o line 352: This phase last around 30s which slightly varies according to the number of symbols.
o line 353: None of the symbol …
o “a custom-made software (SW) that manage”
o “previous experience in the use of BCI”

Response: These errors have been amended in the new version of the document (highlighted in red).

Reviewer 2 Report

To improve the response of ERP components, this paper studied the activation level of ERP and performance of P300-based BCI system from the views of stimulus type and number. Compared with the traditional stimulus, cartoon face stimulus could elicit stronger ERP responses and thus improve the BCI performance. But the number of stimulus symbols did not have significant affects on the ERP responses. This study is done in complete and the results are detailed, which has some reference meaning to BCI design. However, some issues should be addressed before publication:

 1.The section sequence of the manuscript was not consistent with the Introduction part saying ‘The rest of the manuscript is organized as follows: section 2 describes the experimental protocol and the methodology; section 3 presents the results; section 4 discuss the results and presents the conclusions.’

  1. Some abbreviations (e.g. 'SF' and 'CF') appeared without full names at their first times.
  2. As shown in FIG. 7, why use different sizes of stimulus for different types of stimulus? The size of the stimulus would also affect the ERP shape. Pls explain.
  3. When studying ERP responses, the analysis of time-domain signal-to-noise ratio should be added.
  4. Due to the imbalance of sample categories in P300 paradigm, the performance analysis based on the classification accuracy is not complete. So balanced accuracy and AUC should be added.
  5. Only one classification algorithm was used in this paper. The authors are encouraged to compare their work with other algorithms. Some relevant studies should be discussed, such as ‘Discriminative canonical pattern matching for single-trial classification of ERP components[J]. IEEE Transactions on Biomedical Engineering, 2019.’

Author Response

1, The section sequence of the manuscript was not consistent with the Introduction part saying ‘The rest of the manuscript is organized as follows: section 2 describes the experimental protocol and the methodology; section 3 presents the results; section 4 discuss the results and presents the conclusions.’

Response: The indicated errors have been corrected. This error was generated because we adhered to the format that appears in the MDPI (Sensors)’s author guidelines (https://www.mdpi.com/journal/sensors/instructions). Where it is suggested to use the corresponding template in Overleaf for authors of articles (https://www.mdpi.com/authors/latex ) in section “Software and Apps for Writing LaTeX,” subsection “Online with Overleaf.” We also found this structure odd. However, we assumed that it had this order to make the review process more expeditious and so we did not change it. However, the document has already been structured according to the following order: Introduction, Materials and Methods, Results, and Discussion.

2. Some abbreviations (e.g. ‘SF’ and ‘CF’) appeared without full names at their first times.

Response: The indicated errors have been corrected. This error also resulted from not being consistent with the template structure suggested in the guidelines for authors.

3. As shown in FIG. 7, why use different sizes of stimulus for different types of stimulus? The size of the stimulus would also affect the ERP shape. Pls explain.

Response: In the “cartoon face” condition, the phenomenon that produces the evoked response is the substitution of the symbol by a cartoon face. On the other hand, in the “standard flash” condition, the ERP is caused by the contrast of illumination and color of the area surrounding the symbol. There are other works where the symbol has been removed for the standard flash condition, and the evoked potential was observed likewise [Jin (2014)]. Besides, according to the work of Ron-Angevin(2019) and Kellicut-Jones(2018) (both cited in our proposal), the size of stimuli does not influence performance unless the user is unable to control eye movement.

The visual stimuli used in our work guarantee that the user can observe the icon substitution in the cartoon face condition. Additionally, the flashing area in the standard flash is big enough to be perceived by the participants.

4. When studying ERP responses, the analysis of time-domain signal-to-noise ratio should be added.

Response: Thank you for your comment. We have added the signal-to-noise ratio values in captions of Figures 3 and 4 (section “3. Results”, text highlighted in red).

5, Due to the imbalance of sample categories in P300 paradigm, the performance analysis based on the classification accuracy is not complete. So balanced accuracy and AUC should be added.

Response: The accuracy values reported in our work correspond to balanced accuracies. Additionally, we report the recall per class. We erroneously omitted this explanation in the methods of the manuscript. We have added the technical details of how the balanced accuracies were computed. This description is in subsection “2.6 Evaluation process and metrics” of the revised manuscript (text highlighted in red).

On the other hand, we are not including ROC curves because we are not comparing classification models, and we do not have parameters for which we want to obtain the sensitivity and specificity of the machine learning models as they vary. Our study compares the classification rates obtained with the same classification model but different training sets.

6. Only one classification algorithm was used in this paper. The authors are encouraged to compare their work with other algorithms. Some relevant studies should be discussed, such as ‘Discriminative canonical pattern matching for single-trial classification of ERP components[J]. IEEE Transactions on Biomedical Engineering, 2019.’

Response: According to Kellicut-Jones (2018) (cited in our paper), there are two ways to achieve performance improvements in BCIs with ERP paradigms: the first is through manipulation of the paradigm that generates the stimulation; the second is through signal processing techniques, including here the stages of feature extraction and classification. In our work, we are focusing on the first approach. We consider that we do not require a comparison between different classification methods because this research aims to evaluate the model accuracy for different visual stimulation conditions. However, to compare the classification between target and non-target responses with other algorithms is an interesting problem that could be addressed as future work. This has been included in section “4. Discussion”. Also, we included the suggested reference, which provides exciting projections for future work.

Round 2

Reviewer 1 Report

Thank you so much for the rebuttal/clarifications and corrections. I think the paper improved substantially. I still disagree with one or another point but those things seem more personal opinion. You clearly showed the scientific thoughts behind the paper and why it is relevant to your community. I especially see the value in the conducted extensive experiments and providing the respective data.

There are still a few grammatical parts that would benefit from editing but nothing too critical. An English native proof reader is recommended. A few things I detected but there are for sure some more cases. (line 317: shows ->show; line 149: manage->manages; line 335: type->types; line 336: the across->across, line 100: single-option visual stimulation paradigm -> the \emph{single-option} visual stimulation paradigm; line 87: of current work->of this work; The sentence at line 87 is also a bit too long and would benefit from editing. The content is appropriate and important.)

I personally feel that a separate conclusion section might be nice but with the new structuring is not totally required. This is up to the editor.

Some minor points.

In line 271, the term "performance metric" is not appropriate. "Performance evaluation approach" might be a better term. Whereas I see that it is probably too much effort to change the evaluation scheme, for the future, I strongly recommend to use non-randomized 5-fold cross-validation to minimize the contamination. As a better support of your argument in lines 271-272, it would be good to provide a literature reference, where evaluation was done in a similar way.

For the metric discussion in line 276, I recommend that you have a quick look at "Straube S and Krell MM (2014) How to evaluate an agent's behavior to infrequent events?—Reliable performance estimation insensitive to class distribution. Front. Comput. Neurosci. 8:43. doi: 10.3389/fncom.2014.00043 (https://www.frontiersin.org/articles/10.3389/fncom.2014.00043/full) This paper supports your argument on the use of your metric. What I would recommend is to use the term "balanced accuracy" instead of "weighted accuracy" since you are using the special case of a weight of 0.5. Also, "correct classifications for target" should be replaced by "True positive rate" and  "correct classifications for non target" should be replaced by "True negative rate" to be better aligned with the machine learning literature. This is just a recommendation to avoid misunderstandings.

Author Response

There are still a few grammatical parts that would benefit from editing but nothing too critical. An English native proof reader is recommended. A few things I detected but there are for sure some more cases. (line 317: shows ->show; line 149: manage->manages; line 335: type->types; line 336: the across->across, line 100: single-option visual stimulation paradigm -> the \emph{single-option} visual stimulation paradigm; line 87: of current work->of this work; The sentence at line 87 is also a bit too long and would benefit from editing. The content is appropriate and important.)

In line 271, the term "performance metric" is not appropriate. "Performance evaluation approach" might be a better term. Whereas I see that it is probably too much effort to change the evaluation scheme, for the future, I strongly recommend to use non-randomized 5-fold cross-validation to minimize the contamination. As a better support of your argument in lines 271-272, it would be good to provide a literature reference, where evaluation was done in a similar way.

For the metric discussion in line 276, I recommend that you have a quick look at "Straube S and Krell MM (2014) How to evaluate an agent's behavior to infrequent events?—Reliable performance estimation insensitive to class distribution. Front. Comput. Neurosci. 8:43. doi: 10.3389/fncom.2014.00043 (https://www.frontiersin.org/articles/10.3389/fncom.2014.00043/full) This paper supports your argument on the use of your metric. What I would recommend is to use the term "balanced accuracy" instead of "weighted accuracy" since you are using the special case of a weight of 0.5. Also, "correct classifications for target" should be replaced by "True positive rate" and "correct classifications for non target" should be replaced by "True negative rate" to be better aligned with the machine learning literature. This is just a recommendation to avoid misunderstandings.

Response: Thank you very much for your comments. Based on your valuable feedback, we have improved our document, specifically in the following areas:

  • The errors identified on lines 87, 100, 149, 317, 335 and 336 were amended.
  • In line 271 the term "performance metric" was replaced by "performance evaluation approach”.
  • The corresponding references to the sentence covering lines 271 and 272 were added. Similarly, the reference recommended by the reviewer in line 276 was included.
  • In line 276 the term "weighted accuracy" was replaced by "balanced accuracy".

All changes are highlighted in red color.

Reviewer 2 Report

No more comments as the authors have addressed all my questions.

Author Response

Response: Thank you very much for your contribution to the improvement of our work